# Prediction of cycloplegic refraction for noninvasive screening of children for refractive error

**Kazuyoshi Magome[1,2], Naoyuki Morishige[1]\*, Akifumi Ueno[1], Taka-Aki Matsui[1], Eiichi Uchio[2]**

**1** Ohshima Eye Hospital, Fukuoka, Fukuoka, Japan, **2** Department of Ophthalmology, Fukuoka University School of Medicine, Fukuoka, Fukuoka, Japan

\* morishig@corneajp.com

**Data Availability Statement:** The data underlying the results presented in the study are available from (https://doi.org/10.6084/m9.figshare.13650038.v1).

## Abstract

Detection of refractive error in children is crucial to avoid amblyopia and its impact on quality of life. We here performed a retrospective study in order to develop prediction models for spherical and cylinder refraction in children. The enrolled 1221 eyes of 617 children were divided into three groups: the development group (710 eyes of 359 children), the validation group (385 eyes of 194 children), and the comparison group (126 eyes of 64 children). We determined noncycloplegic and cycloplegic refraction values by autorefractometry. In addition, several noncycloplegic parameters were assessed with the use of ocular biometry. On the basis of the information obtained from the development group, we developed prediction models for cycloplegic spherical and cylinder refraction in children with the use of stepwise multiple regression analysis. The prediction formulas were validated by their application to the validation group. The similarity of noncycloplegic and predicted refraction to cycloplegic refraction in individual eyes was evaluated in the comparison group. Application of the developed prediction models for spherical and cylinder refraction to the validation group revealed that predicted refraction was significantly correlated with measured values for cycloplegic spherical refraction ($R = 0.961$, $P < 0.001$) or cylinder refraction ($R = 0.894$, $P < 0.001$). Comparison of noncycloplegic, cycloplegic, and predicted refraction in the comparison group revealed that cycloplegic spherical refraction did not differ significantly from predicted refraction but was significantly different from noncycloplegic refraction, whereas cycloplegic cylinder refraction did not differ significantly from predicted or noncycloplegic values. Our prediction models based on ocular biometry provide estimates of refraction in children similar to measured cycloplegic spherical and cylinder refraction values without the application of cycloplegic eyedrops.

## Introduction

Amblyopia is the most common but treatable pediatric ophthalmologic problem [1]. Early detection of amblyopia allows initiation of treatment at a younger age and consequent normal development of visual acuity in children [2–7]. Screening for refractive error underlying amblyopia in young patients requires the performance of autorefractometry under the

**Funding:** The authors received no specific funding for this work.

**Competing interests:** The authors have declared that no competing interests exist.

cycloplegic condition as a result of the wide range of accommodation in children. However, in the clinical setting, it is often problematic to apply cycloplegic eyedrops to children because of their refusal or the refusal of their parents or of contraindications for the eyedrops [8–10]. Furthermore, the effects of such eyedrops can persist for several hours [11], resulting in disturbance of daily life during accommodation paralysis. Attempts to detect refractive errors in children without application of cycloplegic eyedrops have included the development of procedures for prediction of cycloplegic refraction. The refractive value in children can be estimated on the basis of the ratio of the keratometric value to ocular axial length [12–19]. However, such evaluations rely on spherical equivalent values, with the resultant findings therefore possibly including cylinder refraction, and they may fail to detect the precise refractive error underlying amblyopia. New approaches are therefore needed to determine cycloplegic spherical and cycloplegic cylinder refraction in a screening procedure for amblyopia without the use of cycloplegic eyedrops.

Optical biometry has been applied worldwide to calculate intraocular lens power [20, 21]. It relies on optical coherence to detect various ocular characteristics, and it allows the measurement of ocular axial length as well as corneal anterior curvature, anterior chamber depth (distance from the top of the cornea to the anterior capsule of the lens), lens thickness, and corneal thickness [22–25]. Furthermore, the latest optical biometric techniques based on Fourier domain interferometry minimize measurement times, thereby improving the reliability of the measurements [24, 26]. Such shorter examination times are especially important for measurements in children. Indeed, the application of optical biometry to children has been found to provide reliable measurements of ocular parameters [27, 28], and we therefore anticipated that it might be applicable to assessment of refraction in such young patients without the administration of cycloplegic eyedrops.

Under the noncycloplegic condition, the effect of accommodation in ocular examinations of children cannot be ignored. We therefore considered that a procedure for prediction of cycloplegic refraction in children would need to be based on several clinical parameters. In this study, we collected data for ocular parameters including corneal refraction, corneal astigmatism, ocular axial length, anterior chamber depth, and lens thickness by optical biometry, and we then developed and evaluated prediction models for ocular refraction in children. We found that these models are able to predict cycloplegic spherical and cycloplegic cylinder refraction on the basis of the noncycloplegic measurements made by ocular biometry. Furthermore, the predicted spherical refraction was closer to cycloplegic spherical refraction than was noncycloplegic spherical refraction. Our models may therefore provide a comfortable refractive screening procedure for children without the use of undesirable cycloplegic eyedrops, and they may thus contribute to the early detection and treatment of amblyopia.

## Subjects and methods

### Subjects

The study design was retrospective. Children aged 2 to 9 years who visited Ohshima Eye Hospital from April 2016 through July 2019 were candidates for this clinical study. Out of these 3321 children, those who met the following criteria were eligible for enrollment in the study: (1) their parents provided consent for the examinations; (2) clinical information was available for before and after cyclopentolate application; (3) the standard deviation of the axial length values obtained by ocular biometry was <0.5%; and (4) the range of the values for ocular refraction measured by autorefractometry was <0.75 D. Candidates with diseases of the cornea, lens, or retina, with infectious diseases, with eyelid or orbit abnormalities, or with a history of ocular surgery were excluded from the study. A total of 1221 eyes of 617 children (266

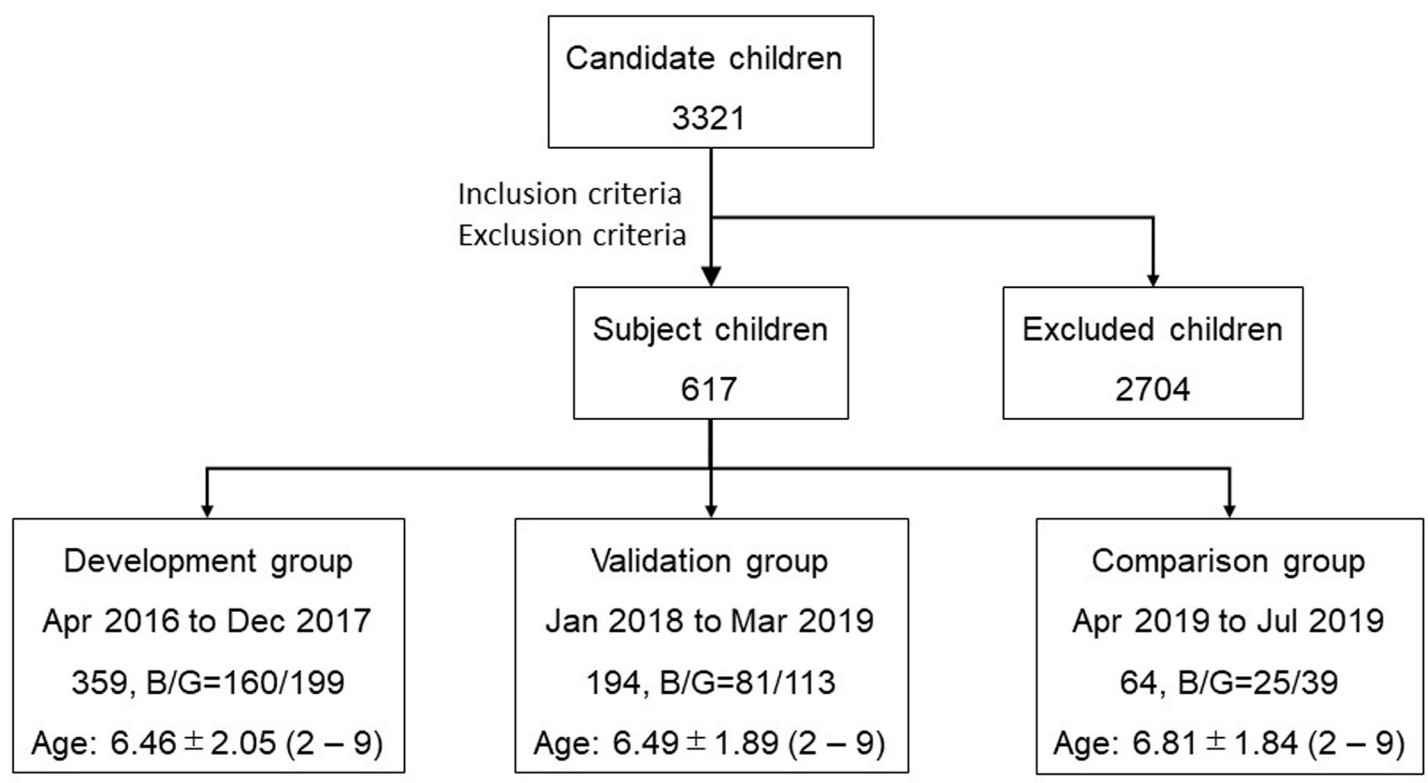

**Fig 1. Flow chart of subject and group selection.** B, boy; G, girl.

boys and 351 girls; mean age ± SD of 6.47 ± 2.00 years, with a range of 2 to 9 years) was eventually enrolled (Fig 1). The subjects were then divided into three groups on the basis of the time period during which they underwent examinations. To develop the prediction models, we selected subjects who were examined from April 2016 through December 2017 as the development group (710 eyes of 359 children; 160 boys and 199 girls, with a mean age ± SD of 6.46 ± 2.05 years and age range of 2 to 9 years). We selected subjects who were examined from January 2018 through March 2019 as a validation group to validate the prediction models (385 eyes of 194 children; 81 boys and 113 girls, with a mean age ± SD of 6.49 ± 1.89 years and age range of 2 to 9 years). Finally, the predicted refraction was compared with measured cycloplegic refraction and noncycloplegic refraction in a comparison group consisting of subjects who were examined from April 2019 through July 2019 (126 eyes of 64 children; 25 boys and 39 girls with a mean age ± SD of 6.81 ± 1.84 years and age range of 2 to 9 years). This retrospective observational study was approved by the Institutional Review Board of Ohshima Eye Hospital (approval no. OEH-2019-03) and adhered to the tenets of the Declaration of Helsinki. Informed consent was obtained from the parents of all subjects.

## Collection of clinical information

All subjects underwent three examinations under noncycloplegic conditions by ocular biometry (IOLMaster 700; Carl Zeiss Meditec AG, Jena, Germany) and by autorefractometry (TONOREFII; NIDEK, Gamagoori, Japan). The examinations were performed by examiners who were unaware of this study. After the noncycloplegic examinations, the subjects received two drops of 1% cyclopentolate (Santen, Osaka, Japan) in each eye with an interval of 5 min between drops. One hour after application of cyclopentolate, loss of direct light reaction and

mydriasis was confirmed and the eyes were examined three times by autorefractometry to obtain cycloplegic spherical and cylinder refraction. The median of all observed values was recorded and used for analysis. The clinical parameters collected by ocular biometry for development of the prediction models were ocular axial length (mm), anterior chamber depth (mm), lens thickness (mm), mean corneal refractive power (D), and corneal astigmatic value (D).

## Development of the prediction models

On the basis of the collected biometric and autorefractometric data, we developed prediction models for spherical refraction and cylinder refraction by applying multiple regression analysis. We first designated axial length [12], anterior chamber depth [29], lens thickness [29], corneal refractive power [12], corneal astigmatism [30], sex [31], and age [32–34] as independent variables, given that they had previously been identified or implicated as factors related to ocular refraction. Cycloplegic spherical refraction and cycloplegic cylinder refraction were considered as dependent (objective) variables. We then applied stepwise multiple regression analysis to identify independent variables that significantly affect prediction of the dependent variables.

## Statistical analysis

We applied stepwise multiple regression analysis to identify statistically significant independent variables for prediction of cycloplegic spherical and cylinder refraction, as described above. The multicollinearity between the selected independent variables had a variance inflation factor of <10. The accuracy of each prediction formula was evaluated by Pearson's correlation coefficient. The Kruskal-Wallis test was applied for multiple comparisons of noncycloplegic refraction, cycloplegic refraction, and predicted refraction in the comparison group. The Dunn-Bonferroni test was applied for comparisons between two groups. A *P* value of <0.05 was considered statistically significant. All statistical analysis was performed with IBM SPSS Statistics software version 24 (IBM, Armonk, NY).

# Results

## Development of the prediction models

With the clinical parameter values obtained from the development group, we developed formulas to predict cycloplegic spherical refraction and cycloplegic cylinder refraction. The statistical method of stepwise multiple regression analysis provides the best combination of suggested independent variables for indication of dependent variables. Such analysis revealed that axial length, anterior chamber depth, lens thickness, corneal refractive power, corneal astigmatism, sex, and age constituted the best combination of independent variables for prediction of spherical refraction (Table 1). Similarly, axial length, corneal refractive power, corneal astigmatism, sex, and age were selected as the best combination of independent variables for prediction of cylinder refraction (Table 2).

On the basis of the stepwise multiple regression analysis, we developed the following multiple regression equation for prediction of cycloplegic spherical refraction:

$$
\begin{aligned}
\text{Predicted spherical refraction} \\
= 96.803 + (-2.588 \times \text{axial length [mm]}) + (-0.985 \times \text{corneal refractive power [D]}) \\
+ (0.285 \times \text{age [years]}) + (1.501 \times \text{anterior chamber depth [mm]}) + (0.377 \\
\times \text{corneal astigmatism [D]}) + (0.453 \times \text{sex [boy}} \\
= 1, \ \text{girl} = 0]) + (-0.479 \times \text{lens thickness [mm]})
\end{aligned}
$$

**Table 1. Stepwise multiple regression analysis for predicted spherical refraction based on clinical information in the development group (710 eyes of 359 subjects).**

| Parameter | Regression coefficient | Standard error | t ratio | P value | Variance inflation factor |
|---|---|---|---|---|---|
| Intercept | 96.803 | 1.756 | 55.116 | <0.001 | |
| Age | 0.285 | 0.021 | 13.533 | <0.001 | 1.719 |
| Sex (boy = 1, girl = 0) | 0.453 | 0.071 | 6.381 | <0.001 | 1.141 |
| Corneal refractive power | −0.985 | 0.026 | −38.528 | <0.001 | 1.263 |
| Corneal astigmatism | 0.377 | 0.037 | 10.169 | <0.001 | 1.088 |
| Axial length | −2.588 | 0.034 | −76.006 | <0.001 | 1.975 |
| Anterior chamber depth | 1.501 | 0.183 | 8.220 | <0.001 | 3.458 |
| Lens thickness | −0.479 | 0.200 | −2.395 | 0.017 | 2.742 |

The correlation coefficient of this multiple regression equation was 0.960 ($R^2$ = 0.922). The error average of the equation was –0.001, and the root mean squared residual (RMSR) was 0.874. Statistical analysis revealed that the $P$ value of this equation was <0.001.

We similarly developed the following multiple regression equation for prediction of cycloplegic cylinder refraction:

Predicted cylinder refraction = –2.877 + (0.911 × corneal astigmatism [D]) + (0.077 × sex [boy = 1, girl = 0]) + (0.055 × axial length [mm]) + (–0.028 × age [years]) + (0.031 × corneal refractive power [D])

The correlation coefficient of the multiple regression equation was 0.844 ($R^2$ = 0.712). The error average of this equation was 0.01, and the RMSR was 0.532. Statistical analysis again revealed that the $P$ value of the multiple regression equation was <0.001.

## Validation of the prediction models

To validate the prediction equations for cycloplegic spherical and cylinder refraction, we applied the clinical information for the validation group to the prediction models and then compared the predicted spherical and cylinder refraction values with the actual cycloplegic spherical and cylinder refraction values for each eye. Fig 2 shows the distribution of cycloplegic spherical refraction and predicted spherical refraction values for the eyes in the validation group. The correlation coefficient of the correlation curve for cycloplegic spherical refraction versus predicted spherical refraction was 0.961 ($R^2$ = 0.924). The error average of the correlation curve was –0.11, and the RMSR was 0.816 (Table 3). Statistical analysis revealed that the $P$ value for the multiple regression equation for cycloplegic spherical refraction and predicted spherical refraction was <0.001. Similarly, the correlation coefficient of the correlation curve for cycloplegic cylinder refraction versus predicted cylinder refraction was 0.894 ($R^2$ = 0.799). The error average of the correlation curve was –0.006, and the RMSR was 0.498 (Fig 3, Table 4). Statistical analysis revealed that the $P$ value for the multiple regression equation for cycloplegic cylinder refraction and predicted cylinder refraction was <0.001.

**Table 2. Stepwise multiple regression analysis for predicted cylinder refraction based on clinical information in the development group (710 eyes of 359 subjects).**

| Parameter | Regression coefficient | Standard error | t ratio | P value | Variance inflation factor |
|---|---|---|---|---|---|
| Intercept | −2.877 | 0.788 | −3.649 | <0.001 | |
| Age | −0.028 | 0.011 | −2.486 | 0.013 | 1.377 |
| Sex (boy = 1, girl = 0) | 0.077 | 0.042 | 1.839 | 0.066 | 1.068 |
| Corneal refractive power | 0.031 | 0.014 | 2.134 | 0.033 | 1.095 |
| Corneal astigmatism | 0.911 | 0.022 | 40.553 | <0.001 | 1.083 |
| Axial length | 0.055 | 0.018 | 3.052 | 0.002 | 1.495 |

### Comparisons among noncycloplegic refraction, cycloplegic refraction, and predicted refraction values

The major and primary procedure used for measurement of spherical and cylinder refraction in the clinic is autorefractometry. We therefore compared cycloplegic measurements obtained

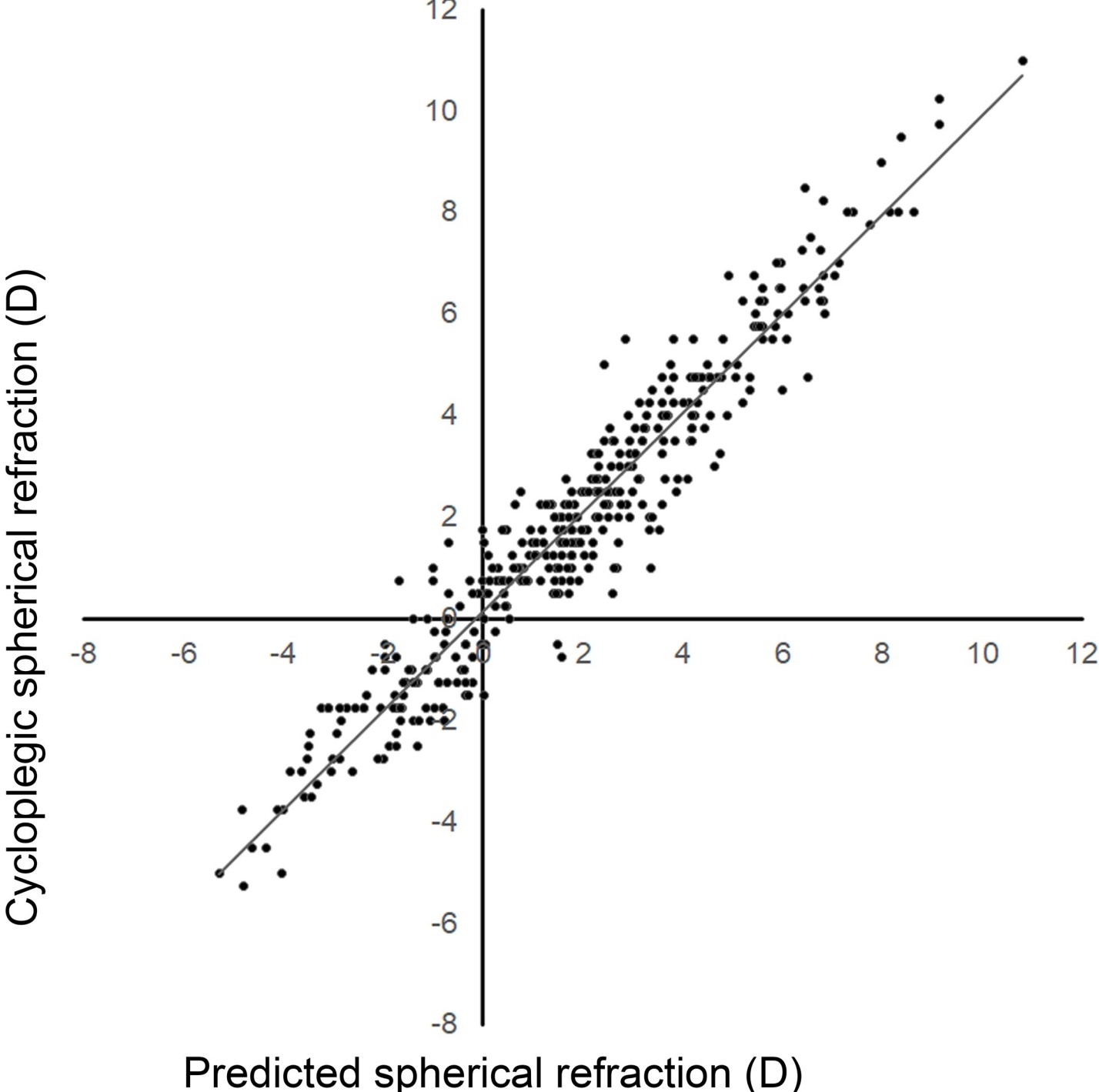

**Fig 2. Distribution of cycloplegic spherical refraction and predicted spherical refraction in the validation group.** Spherical refraction estimated by the prediction formula was highly correlated with cycloplegic spherical refraction ($R^2 = 0.924$).

**Table 3. Validation of model performance for prediction of spherical refraction compared with measured cycloplegic spherical refraction in the validation group (385 eyes of 194 subjects).**

|  | Mean | SD | Minimum | Median | Maximum |
|---|---|---|---|---|---|
| Cycloplegic spherical refraction | 1.81 | 2.94 | −5.25 | 1.75 | 11.00 |
| Predicted spherical refraction | 1.71 | 2.90 | −5.30 | 1.73 | 10.82 |
| Residual error of spherical refraction | −0.11 | 0.81 | −2.66 | 0.14 | 2.36 |

The root mean squared residual was 0.816.

by autorefractometry with noncycloplegic measurements obtained by autorefractometry and with the predicted refraction values obtained by application of clinical parameters measured by ocular biometry for eyes in the comparison group. The correlation coefficient for comparison between cycloplegic spherical refraction and noncycloplegic spherical refraction was 0.929 ($R^2$ = 0.863). The error average of the correlation curve was −0.95, and the RMSR was 1.438 (Table 5). The correlation coefficient for comparison between cycloplegic spherical refraction and predicted spherical refraction was 0.967 ($R^2$ = 0.935). The error average of the correlation curve was −0.12, and the RMSR was 0.727 (Table 5). Similarly, comparison between cycloplegic cylinder refraction and noncycloplegic cylinder refraction yielded a correlation coefficient of 0.980 ($R^2$ = 0.960). The error average of the correlation curve was 0.008, and the RMSR was 0.218 (Table 6). The comparison between cycloplegic cylinder refraction and predicted cylinder refraction revealed a correlation coefficient of 0.910 ($R^2$ = 0.828). The error average of the correlation curve was −0.05, and the RMSR was 0.429 (Table 6). We then compared the distributions of the measured and predicted values of spherical or cylinder refraction. In the case of spherical refraction, the distribution of noncycloplegic measurement values differed significantly from that of cycloplegic measurement values ($P = 0.019$) as well as from that of predicted values ($P = 0.017$), whereas the distributions of cycloplegic measurement values and predicted values were not significantly different (Fig 4). In the case of cylinder refraction, the distributions of cycloplegic measurement values, noncycloplegic measurement values, and predicted values did not differ significantly (Fig 5).

## Discussion

In this study, we developed prediction models for cycloplegic spherical and cylinder refraction in children based on clinical parameters measured by ocular biometry. The prediction equations were found to be precise enough to predict cycloplegic spherical refraction and astigmatism in children. Furthermore, the predicted spherical refraction obtained with our formula was more similar to the measured cycloplegic value than was the noncycloplegic value measured by autorefractometry. Our prediction models are therefore potentially applicable to screening of children for amblyopia and identification of the need for follow-up assessments. They would save time in the clinical evaluation of such children as well as avoid the adverse effects of cyclopentolate eyedrops.

The correlation coefficients for comparisons of predicted spherical refraction with cycloplegic spherical refraction and of predicted cylinder refraction with cycloplegic cylinder refraction in the validation group were 0.961 and 0.894, respectively, suggesting that our prediction models yielded acceptable refractive information in children. It is worthy to note that our formulas estimate cycloplegic refraction values on the basis of ocular biometry data obtained under the noncycloplegic condition. Previous studies have proposed formulas for prediction of refraction values, but the correlation coefficients for comparison of the predicted and measured cycloplegic values ranged from 0.53 to 0.89 [12–19]. The correlation coefficients for

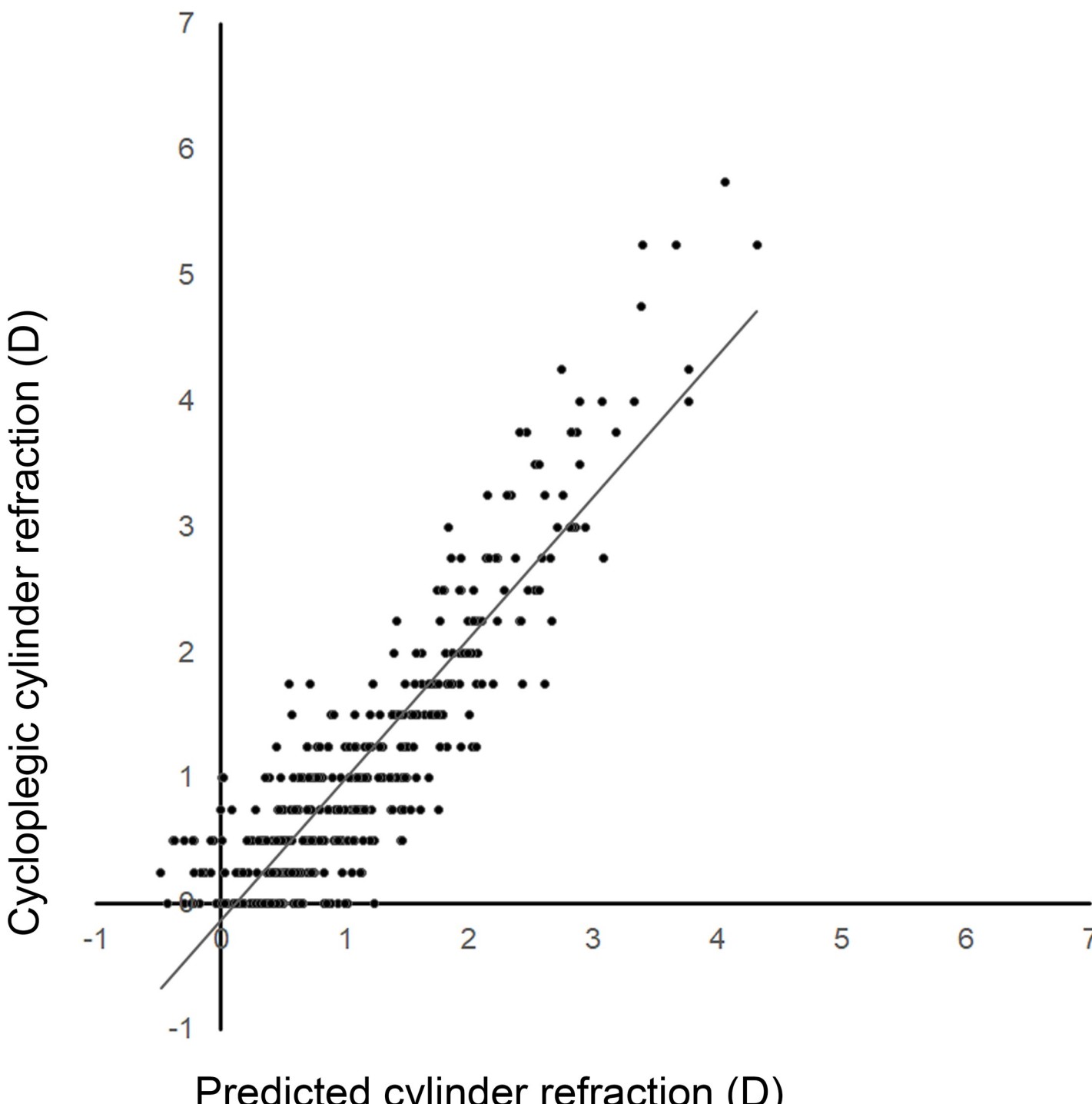

**Fig 3. Distribution of cycloplegic cylinder refraction and predicted cylinder refraction in the validation group.** Cylinder refraction estimated by the prediction formula was correlated with cycloplegic cylinder refraction ($R^2 = 0.799$).

comparisons of predicted and cycloplegic values were 0.53 to 0.81 [12–16] and 0.78 to 0.89 [17–19] in clinical studies of children (3 to 13 years of age) and adults (18 to 39 years of age), respectively. These studies applied clinical information for axial length and corneal curvature.

**Table 4. Validation of model performance for prediction of cylinder refraction compared with measured cycloplegic cylinder refraction in the validation group (385 eyes of 194 subjects).**

|  | Mean | SD | Minimum | Median | Maximum |
|---|---|---|---|---|---|
| Cycloplegic cylinder refraction | 1.13 | 1.09 | 0.00 | 0.75 | 5.75 |
| Predicted cylinder refraction | 1.12 | 0.86 | −0.47 | 0.95 | 4.32 |
| Residual error of cylinder refraction | −0.006 | 0.50 | −1.85 | 0.03 | 1.24 |

The root mean squared residual was 0.498.

The relatively low precision of prediction in young children was speculated to be due to the ongoing development of ocular structure. In comparison with previous studies, our prediction models showed high correlation coefficients between the predicted and cycloplegic values. The application of multiple regression for development of the prediction models, the use of several clinical parameters, and the separate analyses of spherical refraction and astigmatism may all have contributed to the precise prediction of spherical and cylinder refraction in children.

We found that the difference between cycloplegic spherical refraction and either the value predicted by our model or the noncycloplegic value was −0.12 and −0.95, respectively. Younger children show wider accommodation than do adults [35–38]. Furthermore, noncycloplegic autorefraction tends to show a minus overcorrection error in comparison with cycloplegic autorefraction [39–44]. It has thus been difficult to obtain precise measurements of refraction under the noncycloplegic condition in younger individuals as a result of the wider accommodation and myopic shift errors. The predicted spherical refraction provided by our formula was less minus-shifted, and so may be less affected by accommodation in younger children.

On the other hand, our analysis revealed that the difference between cycloplegic cylinder refraction and either cylinder refraction predicted by our formula or noncycloplegic cylinder refraction was −0.05 and 0.008, respectively. Moreover, the correlation coefficients between cycloplegic cylinder refraction and predicted cylinder refraction or noncycloplegic cylinder refraction in the comparison group were 0.910 and 0.980, respectively. Our data thus suggest that measurement of cylinder refraction under the noncycloplegic condition may provide an estimation of cycloplegic cylinder refraction that is as good as or better than that provided by our prediction model, indicating that measurement of astigmatism is less affected by accommodation than is that of spherical refraction.

The collection of refraction data without any effect of accommodation in children requires the application of cyclopentolate eyedrops. The use of cyclopentolate for such cycloplegic evaluation, however, has several disadvantages. Development of the cycloplegic effect requires at least 45 min, and the associated photophobia persists for several hours. Children receiving

**Table 5. Comparison of cycloplegic spherical refraction with noncycloplegic and predicted spherical refraction in the comparison group (126 eyes of 64 subjects).**

|  | Mean | SD | Minimum | Median | Maximum |
|---|---|---|---|---|---|
| Noncycloplegic refractive value | 0.25 | 2.32 | −6.00 | 0.00 | 8.75 |
| Cycloplegic refractive value | 1.20 | 2.82 | −5.75 | 1.25 | 9.75 |
| Predicted refractive value | 1.08 | 2.65 | −5.87 | 1.05 | 9.15 |
| Noncycloplegic vs. cycloplegic |  |  |  |  |  |
| Refractive residual error | −0.95 | 1.08 | −5.50 | −0.50 | 0.25 |
| Predicted vs. cycloplegic |  |  |  |  |  |
| Refractive residual error | −0.12 | 0.72 | −1.95 | −0.15 | 1.51 |

The root mean squared residuals for comparison of noncycloplegic or predicted refraction with cycloplegic refraction were 1.438 and 0.727, respectively.

**Table 6. Comparison of cycloplegic cylinder refraction with noncycloplegic and predicted cylinder refraction in the comparison group (126 eyes of 64 subjects).**

|  | Mean | SD | Minimum | Median | Maximum |
|---|---|---|---|---|---|
| Noncycloplegic cylinder value | 1.07 | 1.08 | 0.00 | 0.75 | 5.50 |
| Cycloplegic cylinder value | 1.05 | 1.02 | 0.00 | 0.75 | 5.25 |
| Predicted cylinder value | 1.01 | 0.86 | −0.30 | 0.75 | 4.21 |
| Noncycloplegic vs. cycloplegic |  |  |  |  |  |
| Cylinder residual error | 0.008 | 0.22 | −0.50 | 0.00 | 0.50 |
| Predicted vs. cycloplegic |  |  |  |  |  |
| Cylinder residual error | −0.05 | 0.43 | −1.36 | 0.04 | 1.24 |

The root mean squared residuals for comparison of noncycloplegic or predicted refraction with cycloplegic refraction were 0.218 and 0.429, respectively.

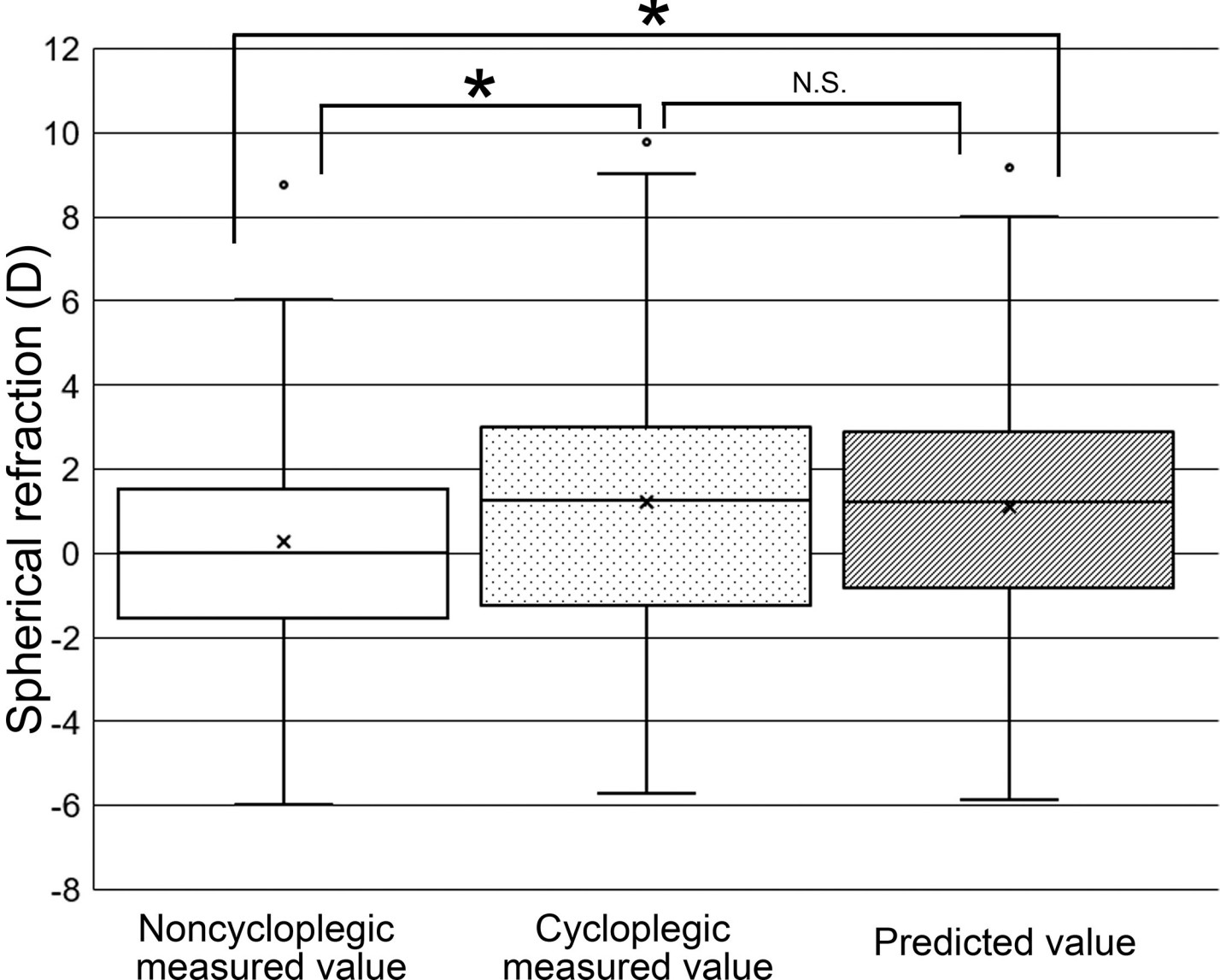

**Fig 4. Box-and-whisker plots showing the distribution of spherical refraction values obtained by noncycloplegic measurement, cycloplegic measurement, and the prediction model.** The top and bottom of each box represent the third and first quartiles, respectively, with the center line representing the median value; whiskers indicate maximum and minimum values; and open circles and crosses denote outliers and the mean, respectively. $^*P < 0.05$; N.S., not significant. The distribution of noncycloplegic measured values was significantly different from that of cycloplegic measured values ($P = 0.019$) and from that of the predicted values ($P = 0.017$). The distribution of cycloplegic measured values was not significantly different from that of the predicted values.

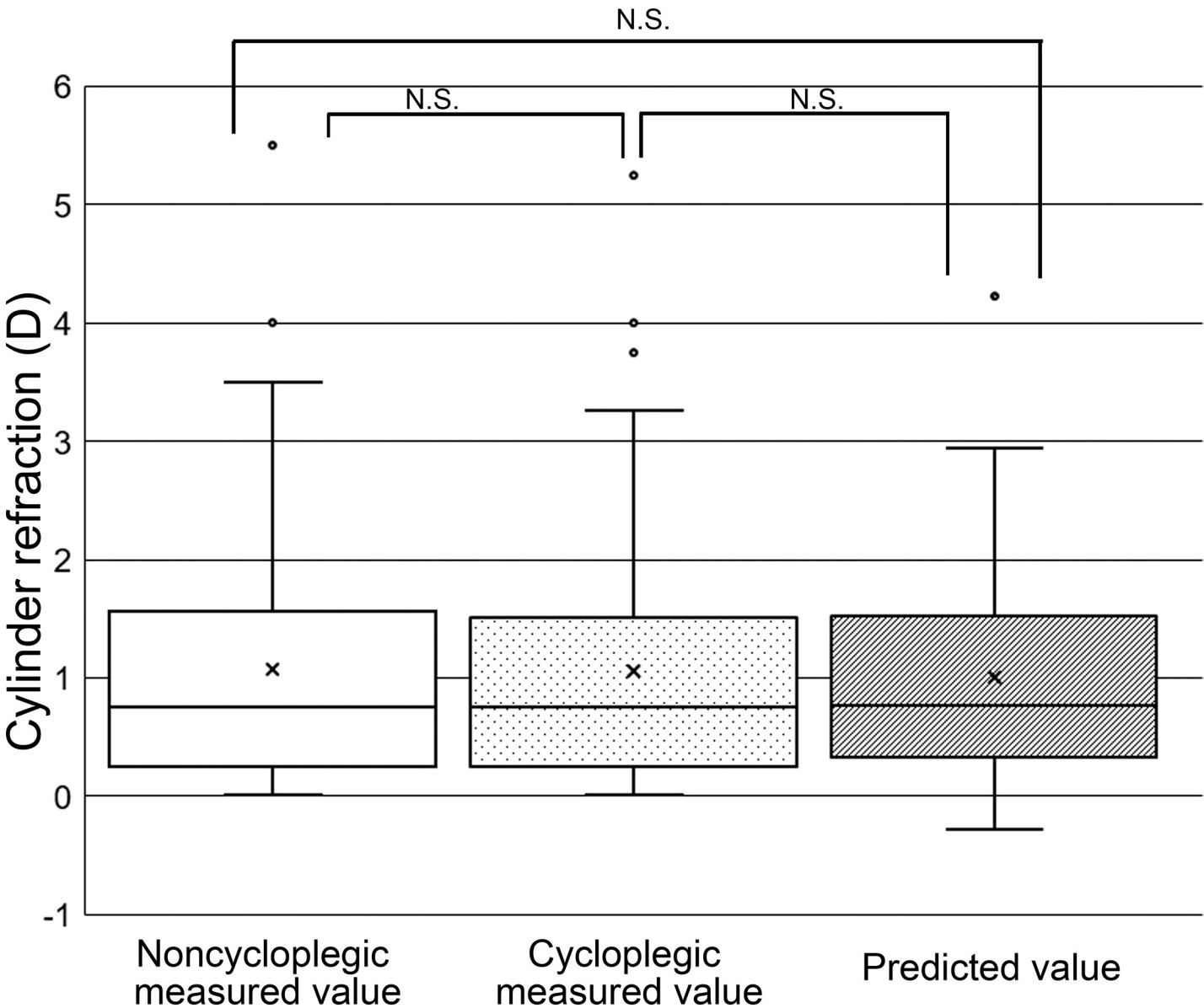

**Fig 5. Box-and-whisker plots showing the distribution of cylinder refraction values obtained by noncycloplegic measurement, cycloplegic measurement, and the prediction model.** N.S., not significant. The distributions of the three groups did not differ significantly from each other.

these eyedrops may also experience ocular irritation, which may result in a lack of patient cooperation. Furthermore, cyclopentolate is associated with a risk for the development of mental disorders or toxicity to the central nervous system [45, 46] or cardiovascular system [47]. These disadvantages sometimes preclude the use of cycloplegic evaluation by application of cyclopentolate eyedrops as a screening procedure to detect refractive errors in children. In the present study, we performed ocular biometry to obtain precise values for axial length, anterior chamber depth, lens thickness, corneal refractive power, and corneal astigmatism within 10 seconds for each eye. Our formulas were developed on the basis of these parameters obtained under the noncycloplegic condition, and they provided predicted refraction values highly similar to the measured cycloplegic refraction values. Thus, with the application of our

formulas, ocular biometric evaluation can provide precise estimates of ocular refraction in children without the adverse effects of cyclopentolate eyedrops and shows potential for screening of children for refractive errors.

However, there are some limitations related to the use of our prediction models. First, there are some challenges to performing ocular biometry in children. The examination requires fixation of the face on the equipment and visual fixation for 5 seconds, which may be difficult to achieve with uncooperative children, especially infants. Second, the astigmatism prediction sometimes yields values less than zero, which may have to be ignored.

In conclusion, we have proposed prediction models based on ocular biometry data for the detection of spherical refractive and astigmatic errors in children. These models provide estimates of spherical and cylinder refraction equivalent to those measured by autorefractometry under the cycloplegic condition while avoiding the risks associated with the use of cyclopentolate eyedrops. Our methodology may also avoid failure to identify anisometropia. We expect that the introduction of our models would allow more accurate noninvasive screening of children for amblyopia and thereby contribute to the prompt management of affected individuals.

## Author Contributions

**Conceptualization:** Kazuyoshi Magome, Naoyuki Morishige.

**Data curation:** Kazuyoshi Magome.

**Formal analysis:** Kazuyoshi Magome.

**Investigation:** Kazuyoshi Magome.

**Methodology:** Eiichi Uchio.

**Supervision:** Akifumi Ueno, Taka-Aki Matsui, Eiichi Uchio.

**Validation:** Akifumi Ueno.

**Writing – original draft:** Kazuyoshi Magome, Naoyuki Morishige, Taka-Aki Matsui, Eiichi Uchio.

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
