## [Decision Letter · Decision Letter 0]

25 Nov 2020

PONE-D-20-33313

Prediction of Cycloplegic Refraction for Noninvasive Screening of Children for Refractive Error

PLOS ONE

Dear Dr. Morishige,

Thank you for submitting your manuscript to PLOS ONE. After careful consideration, we feel that it has merit but does not fully meet PLOS ONE’s publication criteria as it currently stands. Therefore, we invite you to submit a revised version of the manuscript that addresses the points raised during the review process.

ACADEMIC EDITOR:

The manuscript needs grammatical revision. In addition, the study design is not very clear. More details of the study design and the study methods is needed.

q

We look forward to receiving your revised manuscript.

Kind regards,

Ahmed Awadein, MD, Ph.D, FRCS

Academic Editor

PLOS ONE

Journal Requirements:

https://journals.plos.org/plosone/s/file?id=ba62/PLOSOne_formatting_sample_title_authors_affiliations.

Reviewers' comments:

Reviewer's Responses to Questions

**Comments to the Author**

1. Is the manuscript technically sound, and do the data support the conclusions?

Reviewer #1: Yes

Reviewer #2: Yes

2. Has the statistical analysis been performed appropriately and rigorously? 

Reviewer #1: Yes

Reviewer #2: Yes

3. Have the authors made all data underlying the findings in their manuscript fully available?

Reviewer #1: No

Reviewer #2: Yes

4. Is the manuscript presented in an intelligible fashion and written in standard English?

Reviewer #1: Yes

Reviewer #2: Yes

5. Review Comments to the Author

Reviewer #1: Comments:

A welll performed study.

Do you think your prediction models would vary in different ethnic groups considering the high prevalence of myopia in Asia and hyperopia elsewhere?

Can these formulas be incorporated or programmed in ocular biometry machines ?

Line Numbering would have helped greatly with the review.

Grammatical errors:

Abstract:

line 7: "as well as.." replace by" in addition several noncycloplegic parameters were assessed using ocular biometry."

line 12: " were eye" instead of "was"

line 16: "and" instead of 'or"

Introduction:

page 5 line 13: rephrase "precise measurement"

page 6 line 6: "than was measured" replace by " than noncycloplegic..."

Collection of clinical information:

page 8 line 12: remove "the" correct to "under noncycloplegic condition..."

Discussion

page 16: line 15: "of note" replace by " it is worthy to note that "

page 17: line 1: "In particular" needs rephrasing

line 11: remove " by our models"

page 18: line 11: "than is that" change to " than that"

line 15: remove word "thus"

line 16 &17: waht do you mean by "ocular stimulation"?

Page 19: line 14: " however" should be placed at beginning of sentence.

Line 14: Rephrasing needed maybe like " there are some challenges while performing biometery in children"

page 20: line 1: remove word "here".

line 3: provide instead of " provided"

line 4: rephrasing needed maybe to " under cycloplegic condition while avoiding..."

line 5: "or with the failure to ....." start new sentence and rephrase.

Reviewer #2: This study tried to introduce a prediction formula for cycloplegic refraction in children, based on non-cycloplegic ocular biometry parameters. This can avoid the difficulties and the potential hazards encountered with performing cycloplegic refraction, allowing for easier screening and management of amblyopia. The following comments are proposed to improve the work.

Major comments:

1. Abstract: Please mention the study design in the “abstract”.

2. Methods:

• it would be more informative for the reader if you begin the “Methods” section by describing the study design.

• The study design is not clear: Was it totally retrospective? was all participants’ data retrieved by chart review? So, for what purpose did children undergoing refractive error evaluation have routine ocular biometry?

• Page 13: regarding these 2 sentences “Statistical analysis revealed that the P value for the multiple regression equation for cycloplegic spherical refraction and predicted spherical refraction was <0.001.” and “Statistical analysis revealed that the P value for the multiple regression equation for cycloplegic cylinder refraction and predicted cylinder refraction was <0.001”…….What statistical test or parameter did you use for comparison between predicted and measured refraction?

• Regarding comparison group: This group has the same inclusion and exclusion criteria of the validation group. What did this third group add? Why could not such comparison be performed on the validation group?

3. Results: regarding figures: please add a brief description of the figure to figure legends; to allow readers to understand the key message of the figure without referring to the text.

Minor comments:

1. Page 18: “Children receiving these eyedrops may also experience ocular stimulation, which may result in a lack of patient cooperation.” What does “ocular stimulation” refer to?

2. Page 19: “In the present study, we performed ocular biometry to obtain precise values for axial length, anterior chamber depth, lens thickness, corneal refractive power, and corneal astigmatism within 10 s for each eye.” …. “seconds” is better to be written in full.

6. PLOS authors have the option to publish the peer review history of their article (what does this mean?). If published, this will include your full peer review and any attached files.

Reviewer #1: No

Reviewer #2: No

---

## [Author Response · Author response to Decision Letter 0]

26 Feb 2021

Please see the Response to Reviewers file.

---

## [Editor Report · Decision Letter 1]

1 Mar 2021

Prediction of Cycloplegic Refraction for Noninvasive Screening of Children for Refractive Error

PONE-D-20-33313R1

Dear Dr. Morishige,

We’re pleased to inform you that your manuscript has been judged scientifically suitable for publication and will be formally accepted for publication once it meets all outstanding technical requirements.

Kind regards,

Ahmed Awadein, MD, Ph.D, FRCS

Academic Editor

PLOS ONE
---

## [Editor Report · Acceptance letter]

5 Mar 2021

PONE-D-20-33313R1 

Prediction of cycloplegic refraction for noninvasive screening of children for refractive error 

Dear Dr. Morishige:

I'm pleased to inform you that your manuscript has been deemed suitable for publication in PLOS ONE. Congratulations! Your manuscript is now with our production department. 

Kind regards, 

on behalf of

Dr. Ahmed Awadein 

Academic Editor

PLOS ONE